# Calcium supplementation to prevent pre-eclampsia: protocol for an individual participant data meta-analysis, network meta-analysis and health economic evaluation

Thaís Rocha ,[1] John Allotey ,[1] Alfredo Palacios ,[2] Joshua Peter Vogel ,[3] Luc Smits,[4] Guillermo Carroli,[5] Hema Mistry ,[6] Taryn Young,[7,8] Zahida P Qureshi,[9] Gabriela Cormick ,[10] Kym I E Snell,[11] Edgardo Abalos,[5] Juan-Pablo Pena-Rosas,[12] Khalid Saeed Khan ,[13] Koiwah Koi Larbi,[14] Anna Thorson,[12] Mandisa Singata-Madliki,[15] George Justus Hofmeyr,[16] Meghan Bohren ,[17] Richard Riley ,[11] Ana Pilar Betran ,[12] Shakila Thangaratinam,[18,19] On behalf of the International Calcium in Pregnancy (i-CIP) Collaborative Network

**Correspondence to**
Dr John Allotey;
j.allotey.1@bham.ac.uk

## ABSTRACT

**Introduction** Low dietary calcium intake is a risk factor for pre-eclampsia, a major contributor to maternal and perinatal mortality and morbidity worldwide. Calcium supplementation can prevent pre-eclampsia in women with low dietary calcium. However, the optimal dose and timing of calcium supplementation are not known. We plan to undertake an individual participant data (IPD) meta-analysis of randomised trials to determine the effects of various calcium supplementation regimens in preventing pre-eclampsia and its complications and rank these by effectiveness. We also aim to evaluate the cost-effectiveness of calcium supplementation to prevent pre-eclampsia.

**Methods and analysis** We will identify randomised trials on calcium supplementation before and during pregnancy by searching major electronic databases including Embase, CINAHL, MEDLINE, CENTRAL, PubMed, Scopus, AMED, LILACS, POPLINE, AIM, IMSEAR, ClinicalTrials.gov and the WHO International Clinical Trials Registry Platform, without language restrictions, from inception to February 2022. Primary researchers of the identified trials will be invited to join the International Calcium in Pregnancy Collaborative Network and share their IPD. We will check each study's IPD for consistency with the original authors before standardising and harmonising the data. We will perform a series of one-stage and two-stage IPD random-effect meta-analyses to obtain the summary intervention effects on pre-eclampsia with 95% CIs and summary treatment–covariate interactions (maternal risk status, dietary intake, timing of intervention, daily dose of calcium prescribed and total intake of calcium). Heterogeneity will be summarised using tau$^2$, I$^2$ and 95% prediction intervals for effect in a new study. Sensitivity analysis to explore robustness of statistical and clinical assumptions will be carried out. Minor study effects (potential publication bias) will be investigated using funnel plots. A decision analytical model for use in low-income and middle-income countries will assess the cost-effectiveness of calcium supplementation to prevent pre-eclampsia.

**Ethics and dissemination** No ethical approvals are required. We will store the data in a secure repository in an anonymised format. The results will be published in peer-reviewed journals.

**PROSPERO registration number** CRD42021231276.

## STRENGTHS AND LIMITATIONS OF THIS STUDY

⇒ The individual participant data (IPD) approach will allow us to explore any differential treatment effect across groups, and model how individual-level covariates (eg, age, risk of pre-eclampsia) interact with treatment effect within the same trial to explain variability in outcomes.

⇒ By analysing data on the actual amount of calcium taken and adherence to the prescribed regimen, we can explore the doses and frequencies of the clinical benefits of calcium supplementation.

⇒ The health economic analysis will inform decision-makers on current use or future calcium supplementation strategies to prevent pre-eclampsia based on the efficiency principle.

⇒ Limitations include potential unavailability of IPD, which may limit the number of trials included.

## INTRODUCTION

Pre-eclampsia is a pregnancy-specific condition characterised by raised blood pressure and protein in the urine. It is a major cause of maternal and perinatal mortality and morbidity worldwide, contributing to 76 000

maternal and half a million perinatal deaths each year; 99% of these are from low-income and middle-income countries (LMICs).[1–3] Most maternal deaths due to pre-eclampsia are preventable. Prevention of pre-eclampsia and its complications is crucial to achieving the health-related Sustainable Development Goals,[4] and the WHO's Thirteenth General Programme of Work for universal health coverage.[5]

Low dietary calcium is a recognised risk factor for pre-eclampsia.[6–8] In LMICs, 80% of pregnant women have a mean calcium intake below the population Institute of Medicine recommended level of 800 mg/day,[9] compared with low intake in only about a quarter of pregnant women in high-income countries.[10] Calcium supplementation in pregnancy has been shown to reduce the risk of pre-eclampsia.[11] In populations with low dietary calcium intake and in those at high risk of developing pre-eclampsia, the WHO recommends 1.5–2.0 g per day of oral elemental calcium supplementation during pregnancy to reduce the risk of pre-eclampsia, although there is no clear recommendation on the timing of initiation.[12]

A Cochrane review showed that high dose (≥1000 mg per day) of calcium supplementation during pregnancy reduced the risk of pre-eclampsia (8 trials, 10 678 women: average RR 0.36, 95% CI 0.20 to 0.65; $I^2$=76%). But the quality was graded low due to significant heterogeneity from variations in the underlying risk of pre-eclampsia.[11] Evidence for a low-dose calcium supplement to prevent pre-eclampsia (<1000 mg/day) is limited.[11]

Despite countries including calcium in their essential medicines lists, maternal mortality from hypertensive disorders in LMICs remains high.[13 14] Optimising calcium intake to prevent pre-eclampsia is a priority area for the WHO.[5 15] The 2018 WHO Guideline Development Group (GDG) highlighted research on the minimal dose and optimal commencement schedule for calcium supplementation as a high research priority.[15] It is also not known whether calcium supplementation strategies should target high-risk women only or provide calcium supplements to all pregnant and reproductive-aged women, to confer benefits and be cost-effective in preventing pre-eclampsia.

We plan to undertake an individual participant data (IPD) meta-analysis of calcium supplementation to determine the intervention effects on pre-eclampsia and its complications, assess if the effects vary according to maternal and intervention characteristics, and the cost-effectiveness of the different interventions strategies.

## Objectives
Our primary objective is to determine the overall, and differential effects of calcium supplementation (according to maternal and intervention characteristics) on pre-eclampsia adjusted for cointerventions and baseline maternal calcium status, using an IPD meta-analysis.

Our secondary objectives are to:
► Evaluate the effects of calcium supplementation on (1) maternal outcomes such as maternal death, eclampsia, severe maternal morbidity, admission to intensive care unit, haemolysis, elevated liver enzymes, low platelets (HELLP) syndrome and (2) perinatal outcomes such as stillbirth, perinatal death, neonatal death, preterm birth, low Apgar score, small for gestational age baby, and admission and length of stay in the neonatal intensive care unit.
► Produce a rank order of calcium supplementation regimens by effectiveness.
► Develop a decision analytical model to determine the cost-effectiveness of different calcium supplementation strategies in an LMIC setting.

## METHODS AND ANALYSIS
Our IPD meta-analytical approach will follow existing methodological guidelines and adhere to the Preferred Reporting Items for Systematic Review and Meta-Analysis of individual participant data (PRISMA-IPD) reporting statement.[16] The protocol has been registered on the International Prospective Register of Systematic Reviews (PROSPERO; CRD42021231276).

### Patient and public involvement
Women with lived experience of pre-eclampsia will be involved with this work throughout and have informed the design, outcome selection and reporting.

### Literature search
We will update the search of the 2018 Cochrane review[11] until February 2022 to identify new trials that have been published since the last conducted search. This will include searches in databases such as Embase, CINAHL, MEDLINE, CENTRAL, PubMed, Scopus, AMED, LILACS, POPLINE, AIM, IMSEAR, ClinicalTrials.gov and WHO International Clinical Trials Registry Platform, using search strategies adapted from the original Cochrane search, and will include terms for pregnancy such as 'pregnan*' or 'wom*', combined with terms for calcium 'calcium*' and randomised trials 'random*' or 'allocation' (see online supplemental appendix 1). No language restrictions will be applied.

### Eligibility criteria
Any clinical trial with random allocation (individual or cluster) to calcium supplementation (any dose with or without additional supplements or treatments) before or during pregnancy compared with placebo, aspirin or routine care will be eligible for inclusion. Non-randomised trials and animal studies will be excluded.

### Outcome measures
Study outcomes were informed by the WHO recommendation on calcium supplementation during pregnancy to prevent pre-eclampsia and its complications,[17] and the core outcome set for pre-eclampsia research.[18] The primary outcomes are (1) any onset pre-eclampsia and (2) early-onset pre-eclampsia (diagnosed <34 weeks' gestation). We will use the authors' reported definition of pre-eclampsia. However, suppose the trial IPD

| Table 1 | Structured research question |
|---------|------------------------------|
| **Question components** | |
| Population | Pregnant women and women of reproductive age who are not yet pregnant but intending to become pregnant. |
| Intervention | Calcium supplementation (with or without additional supplements or treatments) |
| Outcomes | Primary outcome<br>Any onset pre-eclampsia<br>Early-onset pre-eclampsia (<34 weeks' gestation)<br>Secondary outcomes<br>Maternal outcomes<br>Maternal death<br>Eclampsia<br>Severe maternal morbidity (renal, haematological, neurological, hepatic complications)<br>Admission to intensive care unit<br>HELLP syndrome<br>Neonatal outcomes<br>Stillbirth, neonatal death<br>Apgar score <7 after 5 min<br>Admission to the neonatal intensive care unit within 28 days after birth<br>Preterm birth<br>Small for gestational age baby |
| Design of included studies | Randomised trials |
| HELLP, haemolysis, elevated liver enzymes, low platelets. | |

reports relevant variables. In that case, we will redefine pre-eclampsia as high blood pressure (defined as systolic blood pressure ≥140 mm Hg or diastolic blood pressure ≥90 mm Hg after 20 weeks of pregnancy) with significant proteinuria (defined as urine protein-creatinine ratio ≥30 mg/mmol or ≥2+ on dipstick testing or ≥300 mg/24 hours or ≥500 mg per litre).

Our secondary outcomes include maternal and offspring complications such as maternal death, eclampsia, severe maternal morbidity (renal, haematological, neurological, hepatic complications), admission to intensive care unit, HELLP syndrome, stillbirth, neonatal death, admission and length of stay in the neonatal intensive care unit, preterm birth or small for gestational age (table 1). We will undertake a subgroup analysis to explore whether the intervention effect is modified by (interacts with) maternal risk status, dietary intake, the timing of intervention, the daily dose of calcium prescribed and total intake of calcium.

## Study selection

At least two researchers will independently select studies using a two-stage process. They will first screen the titles and abstracts of studies and then assess the full text of selected studies in detail for eligibility. Disagreement will be resolved via discussion with a third researcher. Data extraction will be done in duplicates. At the study level, extracted data will include country, setting, inclusion and exclusion criteria of participants, intervention, control, primary aim, and definition and assessment of the primary outcome.

## Establishment of the International Calcium in Pregnancy collaborative network

We will contact primary researchers of identified studies via email and invite them to join the collaborative network and share their IPD. To date, seven collaborators have joined the network and shared access to anonymised individual data of 16 111 women (table 2). The network is a global effort to bring together researchers, clinicians and epidemiologists (https://www.icipnetwork.com/). A bespoke database will be set up for collaborators to share data. Authors will be allowed to share their data in any format convenient to them. We will consider all variables recorded in the original studies, even those not reported in the publications. Once deposited, the data will be converted to a standardised format, followed by the range and data consistency checking before merging and harmonising.

## Quality assessment

The quality of the IPD from each study will be assessed independently by two researchers. We will use the revised Cochrane tool for assessing the risk of bias in randomised trials (RoB2)[19] based on published study characteristics and supplement this with information within the IPD. We will consider six items used in the Cochrane risk of bias tool: sequence generation, allocation concealment, blinding, incomplete outcome data, selective outcome reporting and other potential sources of bias. We will conduct sensitivity analyses to examine the robustness of statistical and clinical conclusions to inform the inclusion or exclusion of trials considered to be at high risk of bias.

## Data and integrity checks

We will perform integrity checks of IPD received for each trial by evaluating the integrity of randomisation and follow-up procedures and reviewing the completeness and accuracy of the data.[20] Any inconsistencies found (missing data, extreme values, discrepancies between the trial report and the data) will be resolved with the original study authors. The study progress and discrepancies will be recorded.

## Sample size considerations

Formal sample size calculations are not usually undertaken for meta-analyses. A single trial would need 10 847 participants (80% power, 5% error) to detect the interaction OR of 0.62 between low-risk and high-risk groups, assuming calcium reduces pre-eclampsia by 20% in a low-risk group by another 30% in the high-risk population.[21] Using power calculations by simulating IPD to match aggregate data (eg, number of participants, events,

**Table 2** List of trials current in the i-CIP network and trials that have agreed to share data (total n=17 526 individuals)

| Author, year | Country | Study population risk of PE/start of intervention | Intervention | Comparator | Sample size | Data already shared with the i-CIP network |
|---|---|---|---|---|---|---|
| Trials currently in iCIP (n=16 111 individuals, 7 trials) (data available already) | | | | | | |
| Villar,[49] 2006 | Argentina, Egypt, India, Peru, South Africa, Vietnam | High risk, up to 20 weeks' gestation | 1500 mg calcium carbonate | Placebo | 8325 | Yes |
| Levine,[50] 1997 | USA | Low risk, 13–21 weeks' gestation | 2000 mg calcium carbonate | Placebo | 4589 | Yes |
| Belizán,[51] 1991 | Argentina | Any risk, 20 weeks' gestation | 2000 mg calcium carbonate | Placebo | 1194 | Yes |
| Ettinger,[52] 2009 | Mexico | Low risk, first trimester | 1200 mg calcium carbonate | Placebo | 670 | Yes |
| Goldberg,[53] 2013 | Gambia | Any risk, 18–20 weeks' gestation | 1500 mg calcium carbonate | Placebo | 662 | Yes |
| Hofmeyr,[54] 2019 | Argentina, South Africa, Zimbabwe | High risk, prepregnancy and up to 20 weeks' gestation | 500 mg calcium carbonate | Placebo | 581 | Yes |
| Azami,[55] 2017 | Iran | High risk, >20 weeks' gestation | 800 mg calcium carbonate | Multivitamin | 90 | Yes |
| Trials that agreed to share IPD (n=1415 individuals, 7 trials) (data expected to be made available to us) | | | | | | |
| Omotayo,[56] 2018 | Kenya | Low risk, 16–30 gestational weeks | 1500 mg calcium carbonate | 1000 mg calcium carbonate | 990 | No |
| Asemi,[57] 2014 | Iran | Low risk, 16 weeks' gestation | Multivitamin-mineral with 250 mg calcium | Multivitamin | 104 | No |
| Karamali,[58] 2016 | Iran | High risk, 24–26 weeks' gestation | 1000 mg calcium carbonate, 50 000 IU vitamin D3 | Placebo | 60 | No |
| Samimi,[59] 2016 | Iran | High risk, 20 weeks' gestation | 1000 mg calcium carbonate, 50 000 IU vitamin D3 | Placebo | 60 | No |
| Souza,[60] 2014 | Brazil | High risk, 20–27 weeks' gestation | 2000 mg calcium carbonate, 100 mg aspirin | Placebo | 49 | No |
| Asemi,[61] 2015 | Iran | High risk, 27 weeks' gestation | 800 mg calcium carbonate, 200 mg magnesium, 8 mg zinc, 400 IU vitamin D3 | Placebo | 46 | No |
| Asemi,[62] 2016 | Iran | Low risk, 25 weeks' gestation | 500 mg calcium carbonate, 200 IU vitamin D3 | Placebo | 46 | No |

i-CIP, International Calcium in Pregnancy; PE, pre-eclampsia.

covariate distributions)[22] from studies promising their IPD so far (17 526 women) and assuming heterogeneity of 1%–8% in the rates of pre-eclampsia in the low-risk group in each trial, we will have over 98% power to detect an interaction OR of 0.62 in our IPD meta-analysis.[22] Even when we additionally assume heterogeneity in the overall effect of calcium in the low-risk group from 0.6 to 0.9, the power will still be 90%, illustrating the large sample size available. We will have similar power for other covariates.

## Statistical analysis
### Overall effect

We will perform a series of one-stage and two-stage IPD random-effect meta-analyses fitted using either frequentist methods (eg, restricted maximum likelihood with CIs derived using Hartung-Knapp correction) or Bayesian methods (eg, with vague or empirically derived prior distributions). In the two-stage approach, first, the IPD will be analysed separately for each study to obtain

relevant aggregate data (eg, a treatment effect estimates and its CI for each study) for each outcome; second, this aggregate data will be combined (pooled) across studies using an appropriate meta-analysis model to produce relevant summary results (eg, a weighted average of the treatment effect). The alternative one-stage approach analyses the IPD from all studies in a single step, using a statistical model (eg, a mixed development linear, logistic or Cox regression model) that accounts for the clustering of patients within studies and potential heterogeneity between studies. When the same modelling assumptions and estimation methods are used, one-stage and two-stage approaches are similar.[23] The one-stage approach is preferable when rare events are modelled as a more exact likelihood. However, the two-stage approach allows more familiar meta-analysis techniques and graphs (eg, forest plots). Therefore, we will perform both one-stage and two-stage methods and compare any differences.[23]

### Differential effect by subgroups (treatment–covariate interactions)

For each outcome, we will examine differences in predefined subgroups to summarise whether the intervention effect is modified by (interacts with) maternal risk status, dietary intake, the timing of intervention, a daily dose of calcium prescribed and total intake of calcium; this analysis will use only within-study information to avoid ecological bias from across study information. The one-stage analyses will be achieved by centring patient-level covariates by their mean and including the mean as an additional covariate.[24] Non-linear interactions with continuous covariates (eg, risk status) will be examined using restricted cubic splines.[25]

### IPD network meta-analysis

An IPD network meta-analysis will compare and rank intervention effects for the various regimens (and doses), using direct and indirect comparisons while adjusting for covariates that modify treatment effects to alleviate any inconsistency in the network.[26] The within-study correlation of multiple intervention effects from the same trial will be accounted for (if necessary). A common between-study variance is assumed for all treatment contrasts in the network. We will produce summary (pooled) effect estimates for each treatment contrast (ie, each pair of strategies in the network) with 95% CIs and the borrowing of strength statistics (to reveal the contributions of indirect evidence). Based on the results, the ranking of intervention types will be calculated using resampling methods and quantified by the probabilities of being ranked first, second and last, together with the mean rank and the surface under the cumulative ranking curve. The consistency assumption will be examined for each treatment comparison with direct and indirect evidence (seen as a closed-loop within the network plot); this involves estimating the direct and indirect evidence and comparing the two.[27] The consistency assumption will also be examined across the whole network using 'design-by-treatment interaction' models, which allow an overall significance

test for inconsistency. If evidence of inconsistency is found, explanations will be sought and resolved by adjusting for covariates that act as effect modifiers using the approach of Donegan et al,[28] as identified from the analyses mentioned above.

We will display forest plots for each meta-analysis with study-specific estimates, CIs and weights, alongside the summary (pooled) meta-analysis estimates and a 95% CI. We will translate our findings to the absolute risk prediction scale to help health professionals tailor treatment decisions to an individual's risk of pre-eclampsia conditional on their covariates (prognostic factors) and anticipated treatment effects and any interactions.[29] Penalisation and shrinkage will alleviate overfitting identified using bootstrapping.

### Examining potential sources of bias

Small study effects (potential publication bias) will be investigated using funnel plots and test for asymmetry if ten or more studies are in a meta-analysis. To examine the impact of studies where IPD were not shared, we will extract aggregate study-level data (where available) and incorporate them alongside the IPD using the two-stage random effect meta-analysis framework. We will also examine the impact of excluding any trials that are not at low risk of bias.

### Dealing with missing variables

A range of strategies will be considered for dealing with missing data in covariates. To analyse randomised trials, mean imputation or the missing indicator method are appropriate to handle missing data in covariates.[30] If necessary, we will use multiple imputations for systematically missing variables (considered plausible), which involves borrowing information across studies while allowing for heterogeneity and clustering in a multilevel imputation model.[31]

## Health economic and decision analytical modelling
### Decision model

The cost-effectiveness analysis will be designed and analysed following state-of-the-art methods and analysis in the economic evaluation of healthcare programmes.[32] We will develop a decision tree to determine the cost-effectiveness of calcium supplementation regimens during pregnancy for the prevention of pre-eclampsia. A decision tree is a diagrammatic representation of a decision analysis in which chains of choices are identified, each conditional on a prior choice and with outcomes and probabilities.[33] The model structure will be developed based on previous models.[34–38] The results of the cost-effectiveness analysis will be reported according to the 2022 Consolidated Health Economic Evaluation Reporting Standards statement.[39]

The main outcome of the model will be the incremental cost-effectiveness ratio (ICER). The ICER expresses the additional costs needed to achieve an additional unit of health outcome, that is, the incremental cost per case of

pre-eclampsia/eclampsia (PE/E) avoided. Mathematically, ICER can be expressed as:

$$\frac{Cost_1 - Cost_0}{Health\ benefits_1 - Health\ benefits_0}$$

Where 1 represents the intervention group, and 0 represents the comparator group.

### Intervention and comparators

The interventions to be evaluated (calcium supplementation regimens), as well as their potential comparators, will be defined according to the parent study's 'IPD meta-analysis'.

### Target population

The decision model will be applied to a hypothetical population of pregnant women and women of reproductive age who are not yet pregnant but intend to become pregnant, regardless of their risk for pre-eclampsia and their daily calcium intake. Other populations considered will be pregnant women with a high risk of pre-eclampsia and pregnant women with low calcium intake.

### Study perspective

The study will be conducted from the public healthcare system perspective using IPD estimates for Argentina and published literature.

### Measurement of effectiveness

The health benefits will be measured as cases of PE/E avoided, life-years (LYs) gained and disability-adjusted LYs (DALYs) avoided. For women, we will estimate the LY gained by subtracting the life expectancy from the mean age of an eclampsia patient, while for newborn LY gained will be considered as the average life expectancy in the country. We will use disability weights from the global burden of diseases and country-specific life-expectancy tables for Argentina.[40 41] Results will be presented as cost per case of PE/E avoided, cost per LY gained and cost per DALYs averted.

### Estimating resources and costs

The analysis will also include two main cost categories:

1. Costs of implementing the interventions (calcium acquisition costs, etc).
2. Costs associated with using healthcare services by individuals in both the intervention and comparator groups (hospital stay costs in different complexity of care, laboratory tests, among others). The costs of health events will be estimated for both mother and children using the microcosting method.[42]

### Time horizon

The time horizon will be from pre or early pregnancy until the discharge of mother and child from the hospital.

### Discount rate

Since all costs and PE/E cases will occur within the first year, no discounting will be applied to either cost and PE/E cases. For LY and DALYs, a 3% discount rate will be used in accordance with Bill and Melinda Gates Foundation Reference Case guidelines for LMIC.[43]

### Currency, date, conversions

The costs of implementing the intervention and those associated with the use of healthcare services by individuals will be valued in local currency and then converted to US dollars using international market exchange rates and international dollars through the purchasing power parity conversion factor published by the World Bank database.[44]

### Cost-effectiveness threshold

To define whether the intervention is cost-effective, as the hypothesis is that calcium supplementation will not be 'better and cost saving' than placebo, it will be necessary to establish a decision rule, defined as a willingness-to-pay value for the outcome of interest will be used as a threshold. Despite previous use and recommendations of higher thresholds, such as the WHO's recommendation of up to three times the gross domestic product (GDP) per DALY,[45] we will adopt a more stringent threshold consistent with recent studies: 1 times GDP per capita per DALY or QALY.[46 47] That is, if for a given intervention the ICER lies above this threshold, then it will be deemed too expensive in relation to its added benefit and thus not cost-effective, whereas if the ICER lies below this threshold, the intervention will be judged cost-effective and a 'good buy'. The GDP per capita will be obtained from the World Bank database.[44]

### Sensitivity analysis

Sensitivity analysis will be used to report and assess the level of confidence (or uncertainty) that may be associated with the key model parameters (calcium efficacy, etc). A tornado diagram (deterministic sensitivity analysis) will be generated to plot univariate variations in ICER due to defined variations in key parameters. Probabilistic sensitivity analysis will additionally be performed using 2000 Monte Carlo simulations. We simultaneously sampled from the distributions of each input parameter in each simulation to estimate the 'probability' of the intervention being cost-effective at different thresholds.

### Ethics and dissemination

The current project involves a meta-analysis of anonymised datasets. No ethical approvals are needed for this project. Guidance on participant data storage and management will be adhered to. The dataset is not open access. Findings will be published in peer-reviewed journals, presented at UK national and international conferences, shared with policy-makers and international organisations, and disseminated to women and their families through links with patient groups and relevant charities.

## DISCUSSION

We propose an IPD meta-analysis of randomised trials to evaluate the effects of calcium supplementation in

preventing pre-eclampsia, its complications, and other maternal and fetal–neonatal complications. We will also use an IPD network meta-analysis to compare and rank intervention effects for the various calcium regimens (and doses). In addition, we will assess the cost-effectiveness of calcium supplementation to prevent pre-eclampsia using a model-based economic evaluation for use in LMIC.

The 2018 GDG update reported that calcium supplementation is likely to increase equity. Universal calcium supplementation is expected to prevent 21 500 maternal deaths each year and reduce maternal DALYs by 620 000.[48] However, the dose and timing of choice for optimal calcium supplementation to prevent pre-eclampsia are not yet known. With access to IPD containing over 15 000 participants, our IPD meta-analysis will have a larger sample size than any individual study trying to identify if a particular subgroup benefits the most from calcium supplementation and determine the effects on rare but important outcomes of early-onset pre-eclampsia (delivery <34 weeks' gestation), stillbirth and perinatal deaths, and complications such as HELLP syndrome. By accessing the data on the actual timing of commencement of the intervention, the amount of calcium taken by individual women and their adherence, we can determine if there is an interaction between the effect of calcium treatment and the exact dose taken by the woman. We can then tailor recommendations to the individual conditional on dose and adherence.

Furthermore, our IPD meta-analysis will allow us to tailor calcium treatment strategies considering treatment effects on individual-level factors (including prognostic factors and treatment–covariate interactions). We can model prognostic factors to predict a women's pre-eclampsia risk better, conditional on prognostic factors and the expected response to calcium treatment. Thus, we will combine baseline risk and treatment response information to guide treatment decisions based on individual-level information.

The WHO GDG also highlighted an overall lack of information on the cost-effectiveness of calcium supplementation in LMICs, which is crucial to plan implementation. Therefore, we will evaluate the cost-effectiveness of different calcium supplementation strategies in the LMICs context. To facilitate the adoption of the economic model, we will provide the model in an open-access format. Other researchers can input their country-specific epidemiological and cost data to determine the cost-effectiveness estimates for their countries.

Potential limitations of this study include our inability to obtain IPD from all identified trials due to no contact with original study author, willingness to share raw data or because access to primary data is no longer available. These will be clearly reported as part of our PRISMA flow diagram and a sensitivity analysis to examine the impact of non-IPD studies will be carried out by incorporating these with the IPD studies. There may also be variations in how variables are reported in the shared IPD, which may limit our ability to assess whether the intervention

effect is modified by these individual-level covariates. We will minimise the above limitation through robust data cleaning and harmonisation procedures.

The findings of this IPD meta-analysis and cost-effectiveness analysis will directly inform guidelines and policy-makers in LMICs. The results will assist healthcare managers, other healthcare service providers and policy-makers make informed decisions regarding the ongoing use of calcium or future calcium supplementation strategies to prevent pre-eclampsia based on the efficiency principle.

**Author affiliations**
[1]WHO Collaborating Centre for Global Women's Health, Institute of Metabolism and Systems Research, University of Birmingham, Birmingham, UK
[2]Health Economics, Institute for Clinical Effectiveness and Health Policy, Buenos Aires, Argentina
[3]Maternal, Child and Adolescent Health Program, Burnet Institute, Melbourne, Victoria, Australia
[4]Department of Epidemiology, Care and Public Health Research Institute, Maastricht University, Maastricht, The Netherlands
[5]Centro Rosarino de Estudios Perinatales (CREP), Rosario, Argentina
[6]Warwick Evidence, University of Warwick, Coventry, UK
[7]Centre for Evidence-Based Health Care, Faculty of Medicine and Health Sciences, Stellenbosch University, Cape Town, South Africa
[8]South African Cochrane Centre, South African Medical Research Council, Cape Town, South Africa
[9]Department of Obstetrics and Gynecology, University of Nairobi, Nairobi, Kenya
[10]Department of Health Technology Assessment and Health Economics, Institute for Clinical Effectiveness and Health Policy, Buenos Aires, Argentina
[11]Centre for Prognosis Research, School of Medicine, Keele University, Keele, UK
[12]Reproductive Health and Research, World Health Organization, Geneva, Switzerland
[13]Public Health, University of Granada Faculty of Medicine, Granada, Spain
[14]Action on Preeclampsia (APEC), Accra, Ghana
[15]Effective Care Research Unit (ECRU), East London Hospital Complex, East London, South Africa
[16]Obstetrics and Gynaecology, Frere Hospital, East London, South Africa
[17]Centre for Health Equity, University of Melbourne School of Population and Global Health, Carlton, Victoria, Australia
[18]WHO Collaborating Centre for Global Women's Health, Institute of Metabolism and Systems Research, University of Birmingham College of Medical and Dental Sciences, Birmingham, UK
[19]Birmingham Women's and Children's Hospitals NHS Foundation Trust, Birmingham, UK

**Collaborators** Helen Moraa; University of Nairobi, Rana Zahroh; University of Melbourne

**Contributors** ST and JA planned the study. TR wrote the initial draft of the protocol manuscript with additional input writing input from AP, JPV, LS, KIES, EA, J-PP-R, KSK, KKL, AT, MS-M, RR, GJH, GCa, APB, HM, MB, TY, ZPQ and GCo. TR and JA designed the tables. All authors contributed to the drafts and final version of the manuscript. JA and ST are guarantors. The corresponding author attests that all listed authors meet authorship criteria and that no others meeting the criteria have been omitted.

**Funding** The UKRI Medical Research Council supports this work—Global Maternal and Neonatal Health grant number MR/T010185/1. This work is also funded by the UNDP/UNFPA/UNICEF/WHO/World Bank Special Programme of Research, Development and Research Training in Human Reproduction (HRP), Department of Sexual and Reproductive Health and Research (SRH), WHO. JPV is supported by the NHMRC Investigator grant.

**Disclaimer** The author is a staff member of the World Health Organization. The author alone is responsible for the views expressed in this publication and they

do not necessarily represent the views, decisions or policies of the World Health Organization.

**Competing interests** None declared.

**Patient and public involvement** Patients and/or the public were involved in the design, or conduct, or reporting, or dissemination plans of this research. Refer to the Methods section for further details.

**Patient consent for publication** Not applicable.

**Provenance and peer review** Not commissioned; externally peer reviewed.

**ORCID iDs**
Thaís Rocha http://orcid.org/0000-0003-0113-6877
John Allotey http://orcid.org/0000-0003-4134-6246
Alfredo Palacios http://orcid.org/0000-0001-7684-0880
Joshua Peter Vogel http://orcid.org/0000-0002-3214-7096
Hema Mistry http://orcid.org/0000-0002-5023-1160
Gabriela Cormick http://orcid.org/0000-0001-7958-7358
Khalid Saeed Khan http://orcid.org/0000-0001-5084-7312
Meghan Bohren http://orcid.org/0000-0002-4179-4682
Richard Riley http://orcid.org/0000-0001-8699-0735
Ana Pilar Betran http://orcid.org/0000-0002-5631-5883

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
