## [Reviewer comments · BMJ Open]

ARTICLE DETAILS

TITLE (PROVISIONAL)	Calcium supplementation to prevent pre-eclampsia: protocol for an individual participant data meta-analysis, network meta-analysis, and health economic evaluation
AUTHORS	Rocha, Thaís; Allotey, John; Palacios, Alfredo; Vogel, Joshua; Smits, Luc; Carroli, Guillermo; Mistry, Hema; Young, Taryn; Qureshi, Zahida; Cormick, G; Snell, Kym IE; Abalos, E; Pena-Rosas, Juan-Pablo; Khan, Khalid; Larbi, Koiwah Koi; Thorson, Anna; Singata-Madliki, Mandisa; Hofmeyr, G Justus; Bohren, Meghan; Riley, Richard; Betran, Ana Pilar; Thangaratinam, Shakila

VERSION 1 – REVIEW

REVIEWER	Okoror, Collins E. M. Royal Berkshire Hospital
REVIEW RETURNED	25-Sep-2022

GENERAL COMMENTS	The protocol has been well written, and methodology clearly described. I have made some comments on the background. 'The reviewer provided a marked copy with additional comments. Please contact the publisher for full details.'
--

REVIEWER	Pokhrel, Khem Narayan Liverpool School of Tropical Medicine
REVIEW RETURNED	01-Dec-2022

GENERAL COMMENTS	Thank you for giving the opportunity to review the study protocol: Calcium supplementation to prevent preeclampsia in low- and middle-income countries – individual participant data (IPD) meta-analysis, network metanalysis, and health economic evaluation. I have a few comments here: Title: Why LMIC only? Is it possible to see the data from HIC and compare among the countries? Abstract: Please add sensitivity analysis in the methods section Limitation: One of the major limitations would be Compliance. As it is the major issue of the supplementation, how the analysis is going overcome this limitation. Patient and Public Involvement: This is a review study. Do we take the women in this study, anyway? May be need mention “Not Applicable”.
--

	Literature search: The question is that there is already WHO recommendation for supplementation and supplementation is the policy of most of the LMICs. Not sure if there have been additional trials for these review questions Outcome measures: How the researchers would estimate the dietary intake of calcium, it is quite challenging. Please describe. Sub-group analysis by dietary calcium intake would worth here. Population: Why women of reproductive age are taken here? who are not yet pregnant but intending to become pregnant. Our primary outcome is pre-eclampsia, right? The study wanted to see the pregnancy induced hypertension.
--	--

VERSION 1 – AUTHOR RESPONSE

Reviewer #1 comments

10. The protocol has been well written, and methodology clearly described. I have made some comments on the background.

We thank the reviewer for this comment.

11. Background: Kindly specify the level of significant proteinuria that qualifies for preeclampsia. The presence of any level of protein in urine does not make if preeclampsia.

We stated in the main text of our manuscript how we define significant proteinuria for the purpose of redefining pre-eclampsia if necessary.

“...with significant proteinuria (defined as urine protein-creatinine ratio ≥ 30 mg/mmol or $\geq 2+$ on dipstick testing or ≥ 300 mg/24 hours or ≥ 500 mg per litre)” (Page 8, para 3)

12. Kindly insert reference (for “In LMICs, 80% of pregnant women have a mean calcium intake below the recommended level of 800mg/day”). The WHO & FAO recommend dietary intake of 1200mg/day and the work by Hofmeyr et al did not recommend a level of daily intake.

The Institute of Medicine recommends an estimated average requirement of 800mg of calcium/day as the cut-off point to assess adequacy of calcium intake in a population.[1] This is different from the WHO recommendation of 1.5–2.0g /day of oral elemental calcium supplementation during pregnancy in populations with low dietary calcium intake to reduce the risk of pre-eclampsia.

We have made this clearer in our introduction and provided a reference.

“In LMICs, 80% of pregnant women have a mean calcium intake below the population Institute of Medicine recommended level of 800mg/day, compared with...” (Page 5, para 2)

Reviewer #2 comments

13. Title: Why LMIC only? Is it possible to see the data from HIC and compare among the countries.

The focus of our study as approved by our funder and peer reviewers of our grant proposal is to provide evidence of calcium supplementation to prevent pre-eclampsia in low- and middle-income countries where the greatest disease burden lies. Our IPD meta-analysis will however consider all randomised trials on calcium supplementation from both high income and low-and middle-income countries to inform this. We have updated our title to address this.

14. Abstract: Please add sensitivity analysis in the methods section

We have added the below to the method section of the abstract

“Sensitivity analysis to explore robustness of statistical and clinical assumptions will be carried out.”
(Page 3, Methods and analysis)

15. Limitation: One of the major limitations would be Compliance. As it is the major issue of the supplementation, how the analysis is going overcome this limitation.

Our analysis will consider the actual amount of calcium supplement taken by each individual woman and their adherence, where provided by the original primary trial. We will then be able to determine if there is an interaction between the effect of calcium treatment and its dose or adherence. We can then tailor recommendations to the individual's conditional on dose and adherence.

We stated in our Discussion:

“By accessing the data on the actual timing of commencement of the intervention, the amount of calcium taken by individual women, and their adherence, we can determine if there is an interaction between the effect of calcium treatment and the exact dose taken by the woman.” (Page 21, para 2)

16. Patient and Public Involvement: This is a review study. Do we take the women in this study, anyway? May be need mention “Not Applicable”.

Although this is a review and IPD meta-analysis, women with lived experience of pre-eclampsia have been involved with this work throughout. They will further inform and contribute to the interpretation and reporting of the findings. As such we have included contribution of patient and public in our protocol.

17. Literature search: The question is that there is already WHO recommendation for supplementation and supplementation is the policy of most of the LMICs. Not sure if there have been additional trials for these review questions.

Our search will update and expand on the 2018 Cochrane review and will include searches of the WHO International Clinical Trials Registry Platform and other relevant databases to identify new trials.

18. Outcome measures: How the researchers would estimate the dietary intake of calcium, it is quite challenging. Please describe. Sub-group analysis by dietary calcium intake would worth here.

We will use data on dietary calcium intake as provided by the original primary trial. If this data is not available, we will use estimates of baseline dietary calcium from an international dietary calcium survey. We have previously stated in our methods

“We will undertake a subgroup analysis to explore whether the intervention effect is modified by (interacts with) maternal risk status, dietary intake, the timing of intervention...” (Page 9, para1)

19. Population: Why women of reproductive age are taken here? who are not yet pregnant but intending to become pregnant. Our primary outcome is pre-eclampsia, right? The study wanted to see the pregnancy induced hypertension.

It is important to evaluate whether calcium supplementation before pregnancy leads to a reduction in risk of pre-eclampsia and its complications. The rationale for assessing pre-pregnancy calcium supplementation is that antecedents of pre-eclampsia such as defective placentation might be modified by early calcium supplementation.

Reference

1. Institute of Medicine of the National Academies. Dietary Reference Intakes for Calcium and Vitamin D. Report brief November 2010. Committee to Review Dietary Reference Intakes for Vitamin D and Calcium. Revised March 2011

VERSION 2 – REVIEW

REVIEWER	Okoror, Collins E. M. Royal Berkshire Hospital
REVIEW RETURNED	08-Apr-2023
GENERAL COMMENTS	The authors have edited the protocol based on the comments.